# ACPLI: Automated Channel Pruning with Learned Importance

## Abstract

Neural network pruning allows for significant reduction of model size and latency. However, most of the current network pruning methods do not consider channel interdependencies and a lot of manual adjustments are required before they can be applied to new network architectures. Moreover, these algorithms are often based on hand-picked, sometimes complicated heuristics and can require thousands of GPU computation hours. In this paper, we introduce a simple neural network pruning and fine-tuning framework that requires no manual heuristics, is highly efficient to train (2-6 times speed up compared to NAS-based competitors) and produces comparable performance. The framework contains 1) an automatic channel detection algorithm that groups the interdependent blocks of channels; 2) a non-iterative pruning algorithm that learns channel importance directly from feature maps while masking the coupled computational blocks using Gumbel-Softmax sampling and 3) a hierarchical knowledge distillation approach to fine-tune the pruned neural networks. We validate our pipeline on ImageNet classification, human segmentation and image denoising, creating lightweight and low latency models, easy to deploy on mobile devices. Using our pruning algorithm and hierarchical knowledge distillation for fine-tuning we are able to prune EfficientNet B0, EfficientNetV2 B0 and MobileNetV2 to 75% of their original FLOPs with no loss of accuracy on ImageNet. We release a set pruned backbones as Keras models - all of them proved beneficial when deployed in other projects.

## 1 Introduction

Efforts directed towards deployment of neural networks on low-performance devices such as mobile phones or TVs, created a demand for smaller and faster models. This has led to advances in neural network compression techniques, which allow us to minimize existing large-scale architectures and adjust them to fit specific hardware requirements. Some techniques have been especially successful in this area. Neural network quantization approaches (Nagel et al., 2021) not only decreased the size of the models, but also enabled us to utilize specialized computing accelerators like DSPs. Unfortunately, other techniques, such as network pruning (Liu et al., 2020), are not equally effective in low-resource environments.

Early attempts of naive weight pruning introduced sparse computations, which render them inefficient in practical scenarios (Han et al., 2015; Guo et al., 2016). Channel pruning (Li et al., 2016; Liu et al., 2017; 2021a; Herrmann et al., 2020; Liu et al., 2019b) delivers significant improvements in terms of both memory consumption and execution speed, and is the preferred approach if we want to deploy our models on mobile devices.

However, the majority of existing approaches to channel pruning share several drawbacks:

1. Little effort has been made to address channel interdependencies that occur in the majority of the architectures, with Liu et al. (2021a) being a notable exception. Many popular network architectures contain residual connections inspired by ResNet (He et al., 2015). Feature maps added in residual connections must hold the same shapes, which is likely to be violated when channels are removed independently. We refer to channels involved in this kind of dependency as *coupled*. Automating the process of adding pruning logic to the network in consideration of channel interdependencies is extremely important in practical considerations.

2. Most methods require an expensive and time-consuming fine-tuning process after channels are removed. Some authors use an iterative approach, where channels are removed in a number of steps, and fine-tuning is performed between these steps. Either way, the fine-tuning process often requires a significant number of GPU hours to complete.

3. Channels in any given convolution are being considered independently. However, some target platforms, e.g. SNPE (Qualcomm), are optimized for specific numbers of input and output channels and pruning channels independently can give little to no speed-up.

In order to overcome these issues we introduce an end-to-end channel pruning pipeline which can be deployed on a wide array of neural networks in an automated way. Our main insights are that: (1) Channel

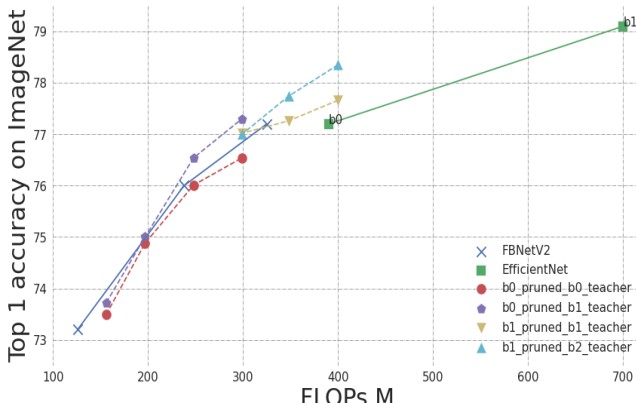

Figure 1: ImageNet accuracy of pruned EfficientNet B0 and B1. Considering FLOPs/accuracy trade-off the some pruned models are better than FBNetV2, which to our knowledge has SOTA results in its FLOPs range.

importance can be learned from the feature maps using simple additional networks and no hand-crafted channel importance metric is needed. (2) Neural network computational graphs should be partitioned in a way which enables removing channels jointly if they are coupled, e.g., if they belong to convolutions whose outputs are later added (the case in point being skip connections). (3) Hierarchical knowledge distillation (a variant of classical knowledge distillation in which multiple teacher networks are used consecutively) is the preferred way of fine-tuning networks after channels are removed since it significantly speeds up training, results in better accuracy and can be used with little or no data augmentation.

Our contributions can be summarized as follows:

1. **New pruning algorithm.** We introduce a new pruning algorithm in which channel importance is learned directly from the feature maps. The process of choosing channels is one-shot and requires just a couple of GPU hours.

2. **Automated method for grouping operations that should be pruned jointly, based on channel interdependencies.** We introduce a relatively simple way of inserting the pruning logic into networks which allows to discard the error-prone process of manual inspection. This makes our solution easy to scale and be deployed for segmentation, detection or image denoising.

3. **Hierarchical knowledge distillation** We employ a novel approach to fine-tuning pruned networks. The approach is related to the one presented in Mirzadeh et al. (2020). The major insight is to gradually increase the complexity of the teacher network. It leads to much quicker training, and yet, achieves much better final accuracy, while not requiring advanced and time-consuming data augmentation procedures.

4. **Up to 25% reduction if FLOPs with no loss in accuracy on ImageNet.** We validate our channel pruning pipeline on:

   - EfficientNet (Tan & Le, 2019), MobileNetV2 (Sandler et al., 2018) and EfficientNetV2 (Tan & Le, 2021) models for classification on ImageNet;
   - a human segmentation model based on EfficientNet B0 with EfficientDet - like Tan et al. (2019) segmentation head;
   - PMRID (Wang et al., 2020) network for RawRGB image denoising.

   In Figure 1 we can see that for FLOPs between 200M and 300M our pruned models outperform FBNetV2 which used, among other things, superkernels to modify existing EfficientNet architecture and produced models that outperform EfficientNet itself. This shows that our pruning and fine-tuning pipeline (which is much simpler than the NAS algorithms used in Wan et al. (2020)) can generate better results. Moreover, EfficientNet B0 pruned to 75% of its original FLOPs has the same accuracy as the original model. Interestingly EfficientNet B1 pruned to match EfficientNet B0 outperforms B0 by approximately 1% in top-1 accuracy on ImageNet.

The whole framework of pruning and fine-tuning we introduce in this paper requires little computational resources. The pruning algorithm usually only takes a couple of hours to complete on a single GPU. Using hierarchical knowledge distillation further speeds up the fine-tuning process.

## 2 RELATED WORK

**Channel selection**. Many channel pruning methods employ a greedy approach where channel removal is interleaved with expensive fine-tuning of the network (Luo et al., 2018; Liu et al., 2015; He et al., 2017).

Similar, but a more affordable approach, is to periodically prune channels throughout a single training procedure (Liu et al., 2021a; Guo et al., 2020; Chen et al., 2020). Ye et al. (2020) and Hou et al. (2021) point out flaws in the idea of greedy channel removal and propose to selectively restore channels in the pruned network. Liu et al. (2019b) trains an auxiliary neural network to quickly evaluate pruned networks and select the best one using an evolutionary algorithm. Other methods jointly train a neural network and learn importance scores for its channels using channel gating mechanism. In (Chen et al., 2020), this is achieved by randomly enable and disable channels during each iteration of the training. Gradient descent was used to update the importance scores in Herrmann et al. (2020); Lin et al. (2020); Ye et al. (2020) and is based on the idea for optimizing hyperparameters in neural architecture search in Liu et al. (2019a) and Xie et al. (2018). These gradient-based methods rely on Gumbel-Softmax reparametrization trick (Jang et al., 2016) to enable back-propagating through the gates distribution. Herrmann et al. (2020) proposes a variant of such a method where the logits of the channel gates are trainable parameters, as well as a variant where the logits are produced by an auxiliary neural network that accepts a feature map. Selecting channels based network input introduces an overhead that is unacceptable on resource-limited devices. Our solution contains a similar idea, but we ensured that the auxiliary networks can be safely removed after the training.

**Channel coupling**. The channel coupling pattern occurs in many modern architectures inspired by ResNet (He et al., 2015), such as MobileNet (Sandler et al., 2018), EfficientNet (Tan & Le, 2019; 2021) or FBNet (Wan et al., 2020). Many studies seem to ignore this issue (Herrmann et al., 2020; Lin et al., 2020; Ye et al., 2020); other resolve this issue by manually grouping interdependent layers or providing model-specific heuristics (Shao et al., 2021; Hou et al., 2021; Guo et al., 2020; Liu et al., 2021b). Independently to our efforts, an automated solution for grouping channels has been proposed in Liu et al. (2021a). We propose a similar algorithm (see section 4), and additionally offer an extension for handling concatenations.

**Measuring speed-up**. Many pruning methods are parametrised by a fraction of channels to prune, either globally or per-layer (Lin et al., 2020; Ye et al., 2020; Herrmann et al., 2020). Overall network FLOPs[1] better corresponds to the usual business requirements. In Chen et al. (2020) and Liu et al. (2021a), the maximal FLOPs parameter is included in their stopping criteria and importance scores of channels are adjusted according to their computation cost. Similarly to Guo et al. (2020), we construct a loss function that introduce a penalty for exceeding the provided FLOPs budget and use it as a part differentiable importance optimization.

**Knowledge distillation**. It has been noted that Knowledge distillation can perform poorly when there is a large discrepancy in complexity between student and teacher networks (Cho & Hariharan, 2019). Cho & Hariharan (2019) evaluate a step-wise approach, in which the intermediate teacher networks are trained by distilling knowledge from the original large teacher and then find it ineffective. Mirzadeh et al. (2020) propose using a *teacher assistant* to bridge the complexity gap. Hou et al. (2021) apply knowledge distillation to fine-tune pruned network, but do not address aforementioned issues. We propose an inverted version of the step-wise approach from Cho & Hariharan (2019), and train train our pruned network with increasingly larger teachers. Such chains can be naturally formed for model families like EfficientNet (Tan & Le, 2019) and EfficientNetV2 (Tan & Le, 2021). We also observe that in case of generic knowledge distillation, the final results can be improved by (even slightly) disturbing the student model with channel pruning before starting the distillation.

## 3 PRUNING METHOD

The basic idea behind our channel pruning algorithm is to set up a scheme in which the importance of channels is being learned from the feature maps generated by convolutions in neural networks. We assign each channel a *score* corresponding to its importance that is updated at each training step and used to approximate behavior of the pruned network by appropriate masking (Liu et al., 2017; Herrmann et al., 2020). Similarly to Herrmann et al. (2020) we apply a probabilistic approach where channels in feature maps are masked with samples from random variables with values in $(0, 1)$. This is a continuous relaxation approach to solving a discrete problem. The distributions of these random variables depend on the values of corresponding **logits** (which can be though of as proxies for channel *scores* and have values in $\mathbb{R}$). These **logits** are learned during the pruning stage. More precisely, given a feature map of size $(B, H, W, C)$ ($B$ is batch size, $H$ and $W$ are spatial dimension and $C$ is the number of channels) and a **logits** variable, for each channel separately we sample — using Gumbel-Softmax (Jang et al., 2016) — the random variable parametrized by the corresponding $logit$ in **logits**. We mask the feature map by multiplying it by the sampled values.

We do not consider each feature map individually — instead, we extend our understanding of channels from a single feature map to a series of operations occurring within a network. The intuition is that element-wise operations, like activation functions, propagate channels forward throughout the network, while convolutional layers *consume* their input channels and *create* new ones. Pruning sequential models is trivial but in more complicated cases, like models with residual connections, there exist additional *couplings* between channels, introduced by operations that accept multiple inputs, e.g. element-wise sum, multiplication (Fig. 2). Because

---

[1]a number of floating-point operations

coupled channels must be pruned jointly to ensure valid shapes, we use a single random variable to mask each set of coupled channels (see Section 4 for details about automatic detection of coupled channels).

Although **logits** can be treated as standalone trainable variables, we choose to learn them from the feature maps in a feedback-loop mechanism. This is because the latter approach is faster to train, results in **logits** which (once converted to probabilities) have lower entropy and produces better results. Once we decide on the feature maps from which we will learn the optimal **logits** values, we place simple neural networks called *logit predictor* modules that take these feature maps as inputs. These modules are build of 3x3 depthwise convolution followed by 1x1 convolution and global mean pooling along spatial dimensions. The output output vector of each such module is later used to update the value of the corresponding **logits** variable (using exponential moving average) as in Figure 2.

The masking operations should always be placed just before the convolution operations that absorb the channels (see Figure 2). The placement of *logit predictors* is more involved and in cases more complicated than the relatively simple one presented in Figure 2, we choose to follow a simple heuristic to place them after convolutions with largest kernel sizes.

During the pruning phase we augment the task-specific loss with an auxiliary latency-based loss. It is based on the expected number of FLOPs in the pruned network, which is computed by using all the **logits** we have attached to the network. We train network weights and *logit predictor* modules jointly so that the network can adjust to channels being phased out.

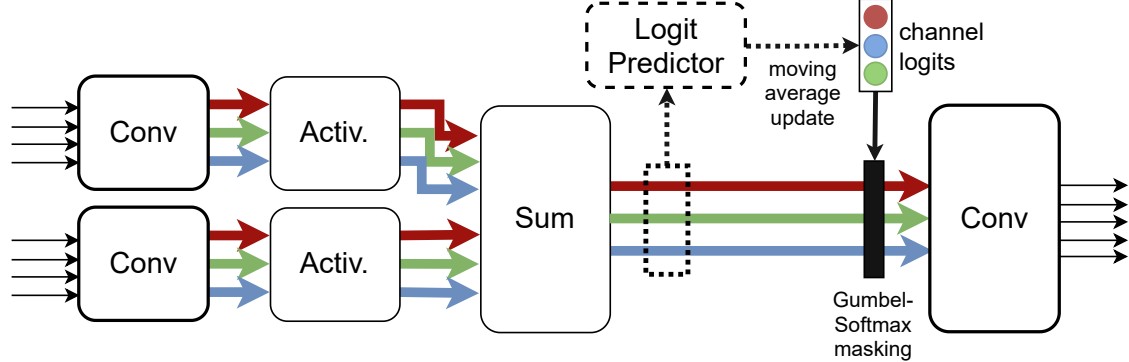

Figure 2: An subset of a network with *logit predictor* and *masking*. The colors indicate the correspondence between channels. The logit predictor takes a feature map produced by the sum operation and use it to predict an update for the channel **logits**.

### 3.1 PRUNING LARGER BLOCKS OF CHANNELS

We allow for blocks of channels (instead of just individual channels) to be treated jointly, so that blocks of a predefined size will be chosen or discarded together. This is especially important for platforms where convolutions are optimized with a specific block size o channels in mind, e.g., for SNPE (Qualcomm) this number is 32 and pruning individual channels often makes little sense.

## 4 LAYER GROUPING ALGORITHM

Although channel coupling has been observed in the literature, relevant groups of operations seem to be usually established via network-specific heuristics or manual annotation. A notable exception is Liu et al. (2021a) where the problem is described at length and an algorithm for finding the groups is derived. The algorithm is then tested on architectures based on ResNet. However, unlike our solution, it does not support concatenation operations. For clarity, we focus on convolutional neural networks, but the proposed strategy can be extended to other kinds of architectures.

### 4.1 SOLUTION

To overcome the issues delineated in Section 3 and make channel pruning available for most off-the-shelf architectures we have developed an algorithm that is capable of automatically detecting channel interdependencies between feature maps generated by operations in the network.

To keep track of all the places where channels have to be considered in a synchronised way, we introduce the concept of an *orbit*. An orbit can be thought as subset of operations that are interdependent from the point of view of channel pruning. Operations in the same orbit need to be considered jointly when removing channels. Naively removing channels without taking into account these interdependencies may result in an invalid network. For example, if we remove an output channel from one of the convolutions on the left in Figure 2,

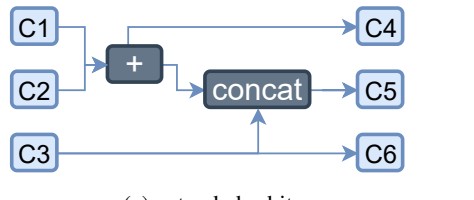 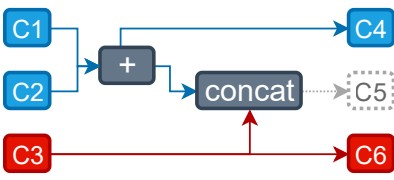

(a) extended orbit       (b) final orbits (red and blue)

Figure 3: Breaking up an extended orbit. An extended orbit is broken up into two final orbits. Nodes $C1$ and $C2$ must have their channels pruned jointly. Node $C3$ can be pruned separately.

the number of channels will no longer match for the *Sum* operation. A typical network has multiple orbits. It is easiest to understand this concept by seeing how orbits are build, which we delineate in Algorithm 1 below.

First, we fix some notation to make matters more intuitive. All the operations in a typical convolutional neural network can be described as being of the following types:

1. **sources** are the operation where new channels are being created, namely regular convolution layers (not depthwise!) and dense layers;

2. **sinks** are the operation where channels are being absorbed, namely regular convolution layers (not depthwise!) and dense layers;

3. **continuators** are all the operations with a single input tensor that simply pass on the channels forward, e.g., batch normalization, mean pooling, resize, activations;

4. **joiners** are operations with multiple input tensors of the same shape which join these tensors without altering the shape, namely element-wise addition and multiplication;

Typically, **continuator** operations are not problematic since they do not alter the channels structure and have a single predecessor and a single output. It is the **joiner** operations that introduce interdependencies between channels. For brevity, from now on we will only speak of convolutions as **sources** and **sinks**, but everything applies just as well to dense layers.

Note that some **sources** can be **sinks** at the same time and vice versa. We refer to operations that are either sinks or sources as **source-sinks**. To identify all the subgraphs in the network where channels have to be considered jointly we run an exhaustive-search type algorithm which has two distinct phases:

**In the fist phase** we search for *extended orbits*, where the coupled operations are brought together. In Algorithm 1 we describe how extended orbits are created. The input is a neural network directed acyclic graph (DAG). The algorithm amounts to removing all inbound edges from convolution nodes and finding all weakly connected components in the resulting graph. The extended orbits are then these weakly connected components once we restore the inbound edges in convolution nodes.

**The second phase** is similar to the first one. For all extended orbits found in phase one we do the following: take the extended orbit and then mark concatenation nodes (which play a special role, since they group channels from separate sources) inside as **sinks** and repeat the process. Most notably, we discard extended orbits in which there are concatenation nodes followed by **joiner** nodes, as it makes the whole process much more difficult to implement. We do not prune channels within such orbits. In Figure 3 we give an example of an *extended orbit* and how is broken up into *final orbits*.

---

**Algorithm 1** Searching for extended orbits

---

    **Input:** network DAG with layers represented as nodes
1:  $\mathcal{P} := \{\mathbf{p} : \mathbf{p}$ is a path starting and ending with a convolution with no convolutions inside the path $\}$
2:  for each path $\mathbf{p}$ in $\mathcal{P}$ remove the last node
3:  for every distinct node $n_i$ on paths in $\mathcal{P}$, create an empty color set for the node $C_{n_i} = \{\}$
4:  $X := \{x : x$ is the initial node of a path in $\mathcal{P}$ $\}$
5:  **for** $x$ in $X$ **do**
6:     pick an unused color $\mathbf{c}$
7:     add color $\mathbf{c}$ to color sets of all the nodes on all the paths in $\mathcal{P}$ starting in $x$
8:  **end for**
9:  **while** there exist nodes with multiple colors **do**
10:    pick a node with multiple colors $\{c_1, c_2, \ldots, c_k\}$ at random
11:    if any node in the DAG has a color in $\{c_2, \ldots, c_k\}$ switch the color to $c_1$
12: **end while**

---

## 5 PRUNING, FINE-TUNING AND HIERARCHICAL KNOWLEDGE DISTILLATION

### 5.1 PRUNING STAGE

The pruning workflow is the same for all types of tasks. We first find all final orbits in the network and attach *logit predictors*. Final orbits determine both: which parts of the network are being pruned and which of them are pruned jointly. The FLOPs per pixel can be automatically computed (and are differentiable with respect to the channel **logits** as in (Fig. 2). We can compute FLOPs for the original network and then set some FLOPs target. In practice we compute $kFPP$ (FLOPs per pixel of the input tensor divided by 1000), to have a value that is independent of the input size. The latency loss is then given by ReLU(kFPP/target_kFPP − 1). We add this loss to the quality loss related to the task, e.g., cross entropy in classification. To avoid an overly aggressive reduction of $kFPP$, we anneal the loss using exponential decay so that at the beginning of training the annealing multiplier is 0. and approaches 1. as the training progresses.

Once the pruning phase is over we retain or discard output channels in convolutions based on channel interdependence discovered by applying Algorithm 1 and the values of **logits** variables learned by *logit predictors*.

### 5.2 FINE-TUNING AND HIERARCHICAL KNOWLEDGE DISTILLATION

We propose to fine-tune pruned models with a method we call *hierarchical knowledge distillation*. This approach relies on increasing the complexity of the teacher network in discrete steps. Given a fine-tuning budget of $K$ GPU hours, and $N$ teacher networks we train the network for $K/N$ GPU hours with each of these teacher networks, starting with the smallest one. Our loss is $L_{ce} + 5L_{kd}$ where $L_{ce}$ is the standard cross entropy loss and $L_{kd}$ is the distillation loss. Using higher weight term for the $L_{kd}$ is crucial to prevent overfitting and produce better results.

Hierarchical knowledge distillation consistently performs much better than just using the original model as the teacher. The comparisons can be seen in Section 6.2. Given an array of models with increasing FLOPs requirements, like EfficientNet Tan & Le (2019) and EfficientNetV2 Tan & Le (2021), it is possible to cheaply train new models for missing FLOPs values. This may produce better results in terms of FLOPs/accuracy trade-off and require less computational resources.

It is perplexing that trying to use *hierarchical knowledge distillation* on an unpruned network does not work anywhere near as well. Our intuition is that pruning provides some kind of initial perturbation to network weights and architecture which prove beneficial from the point of view of gradient descent optimization. Are there any other types of model perturbations which boost the effectiveness of this type of knowledge distillation? These are the questions we could try to address as our future research. It would be also interesting to see how this approach performs when applied to recent state-of-the-art methods based on neural architecture search Wang et al. (2021).

## 6 EXPERIMENTS

All the experiments we perform adhere to the same schedule: (1) We first run the pruning algorithm with additional latency losses (usually 1-10 epochs, depending on the task). (2) We then fine-tune the pruned model (without resetting its weights). The experiments for classification on ImageNet are presented in Section 6.2. Experiments for image denoising and human segmentation are presented in Sections A.2.1 and A.2.2, respectively.

### 6.1 HYPERPARAMETERS FOR THE PRUNING PHASE

For the pruning phase, during which channels to be removed are being chosen, the setup is roughly the same for each task. The *logits predictor* is always a two layer network with $3 \times 3$ depthwise convolution followed by $1 \times 1$ convolution and global mean pooling. We set the batch size to 16 and run the training updating the channel gates distributions as described in section 3. The initial value of channel **logits** is set to 3.0 so that initially there little to no masking. There is an additional loss that penalizes the entropy of all the **logits** so that at the end of the pruning phase the channel enabling probabilities (which we get by applying softmax to **logits**) are far away from 0.5. The temperature for Gumbel-Softmax is constant - 0.5.

### 6.2 CLASSIFICATION ON IMAGENET

We prune EfficientNet B0, EfficientNet B1 (Tan & Le, 2019), MobileNetV2 (Sandler et al., 2018), and EfficientNetV2 (Tan & Le, 2021). We choose these since they are already highly optimized for mobile devices and relatively small. EfficientNetV2 is a recent state-of-the-art architecture optimized for mobile GPUs and DSPs. All the models are taken from their official *Keras* implementations[2] except for EfficientNetV2. Larger

---

[2]https://www.tensorflow.org/api_docs/python/tf/keras/applications

Table 1: Top 1 ImageNet accuracy and FLOPs for for EfficientNet B0 and B1 pruned

(a) B0 pruned

| Model | Standard training | B0 teacher | B1 teacher (after using B0 first) | FLOPs (G) |
|---|---|---|---|---|
| *original* | 77.30 | - | - | 0.393 |
| $m6$ | - | 73.49 | 73.72 | 0.156 |
| $m8$ | - | 74.88 | 75.00 | 0.197 |
| $m10$ | - | 76.01 | 76.56 | 0.248 |
| $m12$ | - | 76.54 | 77.30 | 0.299 |
| NPRR Hou et al. (2021) | - | 77.00 | - | 0.346 |

(b) B1 pruned

| Model | Standard training | B1 teacher | B2 teacher (after using B1 first) | FLOPs (G) |
|---|---|---|---|---|
| *original* | 79.10 | - | - | 0.700 |
| $m12$ | - | 77.02 | 76.99 | 0.299 |
| $m14$ | - | 77.26 | 77.74 | 0.348 |
| $m16$ | - | 77.66 | 78.35 | 0.400 |

networks like the VGG19 or the ResNet family has been predominant in channel pruning literature, but are rarely suitable for resource-limited devices, where the need for optimization is biggest. The phase where channels are chosen usually lasts a little more than a single epoch on ImageNet. We split the ImageNet train data into two parts, leaving about 5% of the data for early-stopping.

Following Section 5.2 we use multiple teacher networks. The details are as follows:

- **EfficientNet B0**: fine-tune the models for 40 epochs with **B0** as teacher and then we further fine-tune with a **B1** for another 40 epochs;

- **EfficientNet B1**: fine-tune the models for 25 epochs with **B1** as teacher and then we further fine-tune with a **B2** for another 25 epochs.

- **MobileNetV2**: fine-tune the models for 40 epochs with **MobileNetV2** as teacher and then we further fine-tune with a **EfficientNet B0** for another 40 epochs.

- **EfficientNetV2 B0**: fine-tune the models for 16 epochs with **B0V2** as, then fine-tune the models for 16 epochs with **B1V2** as teacher and finally fine-tune the models for 16 epochs with **B2V2** as teacher.

The interesting thing we noticed is that using knowledge distillation without pruning does not help at all. For example we tried fine-tuning MobileNetV2 with EfficientNet B0 teacher right away and top 1 Imagenet accuracy fell from $71.52\%$ to $71.12\%$. We conjecture that some kind of initial perturbation is needed for knowledge distillation to work. In our case this perturbation is channel pruning.

Batch size is set to 192 for **B0** and **MobileNetV2** fine-tuning. For **B1** and **EfficientNetV2 B0** batch size is 128. The input image resolution is $(224, 224)$. We use only random crop and flip as augmentations. For training we use one NVidia RTX3090 GPU. For the pruning phase we set the batch size to 16 and, quite importantly, we freeze all batch normalization layers. We use Adam optimizer for all the training runs. During mask-learning phase the learning rate is set to $0.0001$. For fine-tuning we use exponential decay with learning rate initially set to $0.0001$ and the decay rate set to $0.001$.

### 6.2.1 COMPARISONS AND DISCUSSION

Few authors have attempted to prune EfficientNet (Tan & Le, 2019). We can compare our results with Hou et al. (2021), where only one model is presented, which was also fine-tuned with knowledge distillation. We provide a much wider FLOPs spectrum for B0 and prune B1 as well. It is interesting to see that B1 pruned to the FLOPs level of B0 outperforms B0 by a wide margin. The results are in Table 1.

Comparisons for MobileNetV2 are quite difficult due the inconsistencies between different versions of the model taken by different authors as their baseline. For instance in Hou et al. (2021) the authors first take an *over-pruned backbone* which they proceed to prune. In Liu et al. (2019b) the largest version of MobileNetV2 is taken (585M FLOPs) and then pruned. Some of the authors run the fine-tuning for much longer than we do. Notably, in Ye et al. (2020) the fine-tuning is run on 4 GPUs with batch size 512 and for 250 epochs which is considerably more expensive than our approach. Detailed results are in Table 2 and Figure 5a. Again using hierarchical knowledge distillation we are able to fine-tune the model pruned to 75% of original FLOPs so that it has 0.7% higher accuracy than the original.

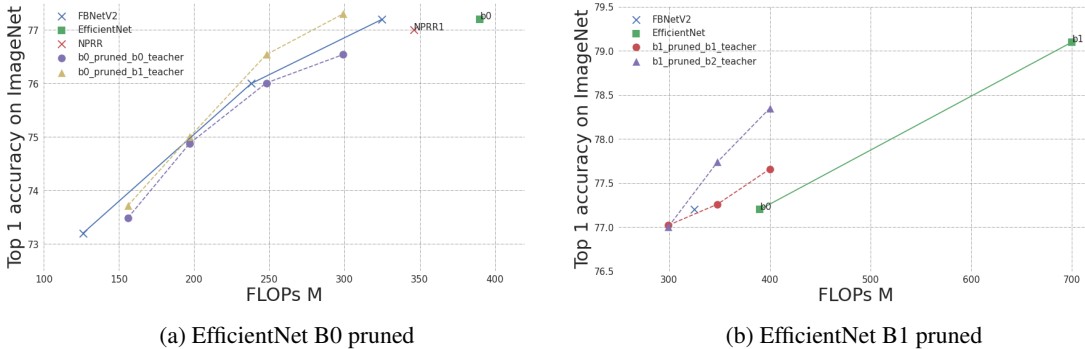

(a) EfficientNet B0 pruned

(b) EfficientNet B1 pruned

Figure 4: ImageNet accuracy of pruned EfficientNet B0 and B1. Considering FLOPs/accuracy trade-off the some pruned models are better than FBNetV2.

Table 2: Top 1 ImageNet accuracy and FLOPs for MobileNetV2 pruned

| Model | Standard training | MobileNetV2 teacher | B0 teacher (after using MobileNetV2 first) | FLOPs (G) |
|---|---|---|---|---|
| *original* | 71.52 | - | - | 0.301 |
| $m5$ | - | 67.58 | 67.99 | 0.135 |
| GSPE Ye et al. (2020) | 68.8 | - | - | 0.138 |
| $m6$ | - | 67.08 | 68.76 | 0.140 |
| META Liu et al. (2019b) | 68.2 | - | - | 0.140 |
| GFP Liu et al. (2021a) | 69.16 | - | - | 0.150 |
| GSPE Ye et al. (2020) | 69.7 | - | - | 0.152 |
| $m7$ | - | 69.79 | 70.05 | 0.170 |
| GSPE Ye et al. (2020) | 70.4 | - | - | 0.170 |
| $m8$ | - | 69.47 | 71.28 | 0.199 |
| GSPE Ye et al. (2020) | 71.2 | - | - | 0.201 |
| GSPE Ye et al. (2020) | 71.6 | - | - | 0.220 |
| $m9$ | - | 70.92 | 72.22 | 0.228 |

When it comes to EfficientNetV2, we are able to outperform the original model's results on ImageNet with the help of hierarchical EKD, inasmuch as the pruned version of B0 (70% of the FLOPs of the original model) has higher top 1 accuracy than the original. See Table 3 and Figure 5b.

# 7 CONCLUSION

Using an automated solution to process coupled channels in neural network architectures and a simple scheme to learn channel importance, we are able to prune models with varying architectures for different underlying tasks. For fine-tuning pruned classification networks we use hierarchical knowledge distillation which produces much better results than just using the original model as a teacher. The whole pruning pipeline requires much less computational resources than some of the state-of-the-art NAS based solutions for finding efficient FLOPs / accuracy trade-offs like Wang et al. (2021).

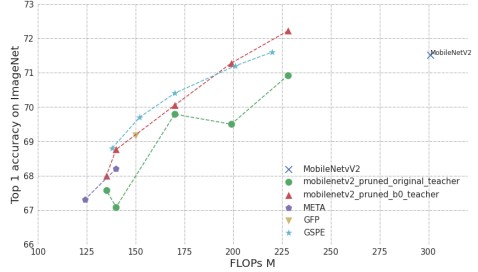

(a) ImageNet accuracy of pruned MobileNetV2. For finetuning we use knowledge-distillation with both original model and EfficientNet B0 as teachers.

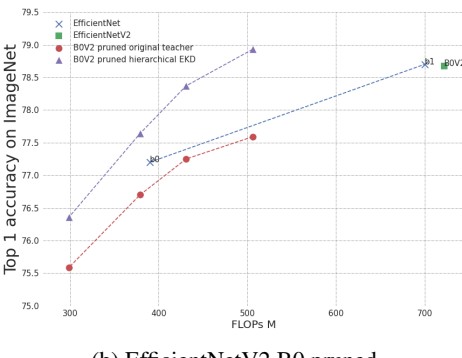

(b) EfficientNetV2 B0 pruned

Figure 5: Pruning results for MobileNetV2 and EfficientNetV2

Table 3: Top 1 ImageNet accuracy and FLOPs for EfficientNetV2 B0 pruned

| Model | Standard training | B0 teacher | hierarchical teachers | FLOPs (G) |
|---|---|---|---|---|
| *original* | 78.67 | - | - | 0.722 |
| $m20$ | - | 77.59 | 78.93 | 0.506 |
| $m17$ | - | 77.25 | 78.37 | 0.431 |
| $m15$ | - | 76.70 | 77.64 | 0.379 |
| $m12$ | - | 75.59 | 76.36 | 0.299 |

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

## A APPENDIX

### A.1 LAYER WIDTHS VISUALIZATION

It is quite interesting to see how layer width looks like after pruning. The pattern that emerge are quite telling. EfficientNets are build of a series of meta-blocks, .e.g, $2, 3, \ldots, 7$ in EfficientNet B0, where each meta-block consists of a number of MBCONV blocks at the same spatial resolution. It appears that in each such meta-block the most important block is usually the first one, and block importance decays proportionally to the depth of the block inside the meta-block. See Figure 6 in the Appendix.

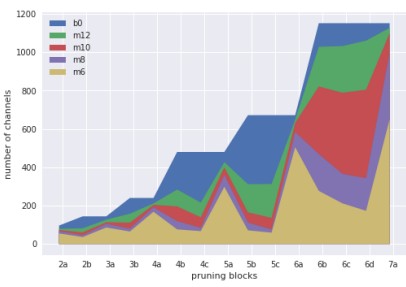
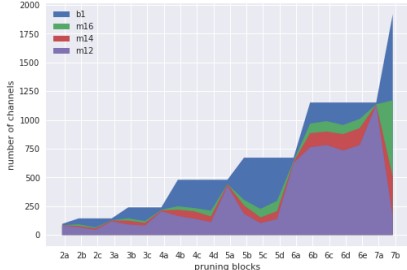

(a) EfficientNet B0 pruned  (b) EfficientNet B1 pruned

Figure 6: Visualisation of the layer width after channels are removed. There is a noticeable patter in which the first block in a series of residual blocks at the same spatial resolution is the most important one and the algorithm is reluctant to remove the channels. Later blocks seem to be less informative, proportionally to their depth.

### A.2 FURTHER RESULTS

#### A.2.1 RAWRGB IMAGE DENOISING

We prune a recent state-of-the-art network for RawRGB image denoising on mobile devices introduced in Wang et al. (2020). We train the models on **SIDD Medium** dataset https://www.eecs.yorku.ca/~kamel/sidd/dataset.php. We first extract 256x256 patches for training and validation and then test the networks on SIDD validation dataset https://www.eecs.yorku.ca/~kamel/sidd/benchmark.php. The batch size is set to 16, learning rate is 0.0001 and we use Adam optimizer. The loss is mean absolute error. We train the original model for 150 epochs, prune it and then train the original model for another 150 epochs. The pruned models are fine-tuned for 150 epochs as well. For comparison we also train from scratch smaller (linearly scaled down) versions of the original model. The results can be seen in Table 4 and Figure 7.

#### A.2.2 HUMAN SEGMENTATION

For semantic segmentation we use a private dataset for training human segmentation models for real time prediction in video bokeh task. This is dictated by the need to have superior edge quality which is missing in publicly available data for segmentation. The dataset consists of 120k real image/mask pair and 50k synthetic ones. Apart from IoU we also compute edge IoU, which pays attention only to the edges of the masks and can be thought of as a proxy for edge quality. The baseline architecture consists of an EfficientNet

Table 4: Pruning results for image denoising and human segmentation.

(a) PSNR and kFPP for pruned PM-RID model

| Model | PSNR | kFPP |
|---|---|---|
| $baseline$ | 51.84 | 29.9 |
| $m25$ | 51.84 | 25.4 |
| $m22$ | 51.79 | 22.1 |
| $m20$ | 51.80 | 19.6 |
| $m15$ | 51.67 | 15.1 |
| $m12$ | 51.41 | 12.3 |

(b) Pruning results for pruned human segmentation model.

| Model | IoU | Edge IoU | kFPP (G) |
|---|---|---|---|
| $baseline$ | 0.9414 | 0.4039 | 40.8 |
| $m30$ | 0.9440 | 0.3977 | 30.3 |
| $m25$ | 0.9423 | 0.3848 | 25.4 |
| $m22$ | 0.9431 | 0.3816 | 22 |
| $m19$ | 0.9420 | 0.3574 | 19.3 |
| $m15$ | 0.9372 | 0.3354 | 14.6 |
| $m11$ | 0.9295 | 3213 | 11.3 |
| $m8$ | 0.9253 | 0.2939 | 7.5 |
| $m5$ | 0.9050 | 0.2265 | 4.5 |

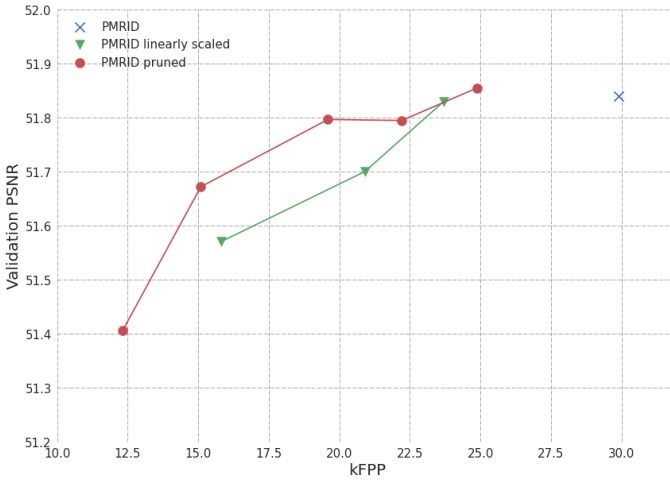

Figure 7: Validation results for pruned RawRGB denoising models.

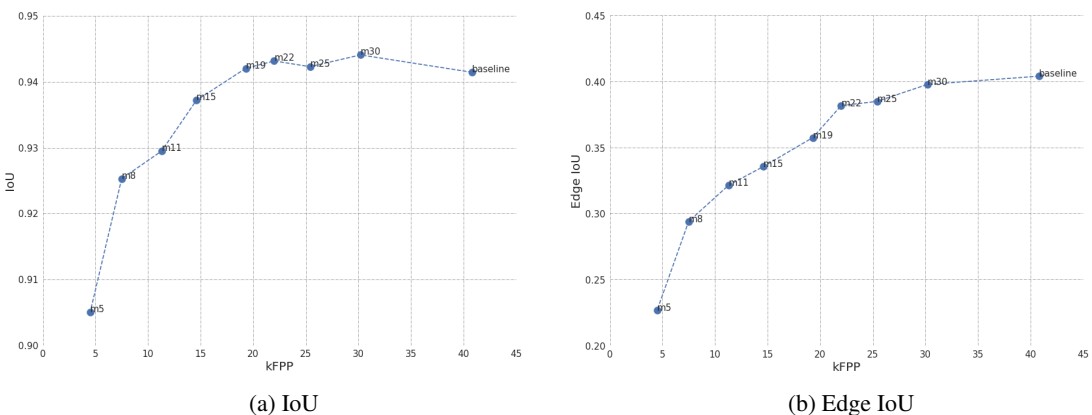

(a) IoU

(b) Edge IoU

Figure 8: Validation results for pruned human segmentation.

B0 (Tan & Le, 2019) backbone, EfficientDet (Tan et al., 2019) (modified slightly to allow for easier channel pruning) fusion block and a detail branch (Siam et al., 2018) to preserve edge quality. The backbone network is pretrained on ImageNet. We train the original model for 70 epochs, prune and then fine-tune the pruned models for 50 epochs. The validation results are presented in Table 4. The validation dataset is a split of a modified version of LIP dataset (Gong et al., 2017), where objects belonging to people (such as handbags, etc.) are also considered part of these people. This is done, so that we can train models for video bokeh effect. The results are in Table 4b and are visualized in Figures 8a and 8b.

Notice that the smallest pruned model is compressed to around 10% of the size of the original one. Even in these extreme compression scenario our approach produces a model with IoU higher than 90%. IoU starts dropping only after we have removed more than 60% of the original FLOPs. This is an observation which, in our experience, is true for many more architectures for segmentation, the one being presented here is just one example. Edge IoU starts falling much more quickly, perhaps beacause we employ no edge-specific loss.

