# OpenReview forum: "Automated Channel Pruning with Learned Importance"
_ICLR.cc/2022/Conference — ICLR 2022 Submitted_

### Official Review · Reviewer_Www2 · 2021-10-25

**Correctness:** 3
**Technical Novelty And Significance:** 3
**Empirical Novelty And Significance:** 2
**Recommendation:** 5
**Confidence:** 5

**Main Review:**

Strengths:
1) The proposed pruning scheme solves the dimension mismatching problem, when pruning some specific model architecture (such as ResNet).
2) The hierarchical KD can promote the performance, which avoids the problem: “too large teacher does not always make better student”.

Weakness:
1)	Although each part of the proposed method is effective, the overall algorithm is still cumbersome. It has multiple stages. In contrast, many of existing pruning methods do not need fine-tuning.
2)	Technical details and formulations are limited. It seems that the main novelty reflected in the scheme or procedure novelty.
3)	The experimental results are not convincing. The compared methods are few. Although few authors have attempted to prune EfficientNet, other networks can be compressed in experiments such as ResNet. In addition, the performance gains compared with SOTAs are also marginal, which are also within 1%.
4)	The paper is poorly written. There are many typos and some are listed as follows:
       --In caption of Figure 2, “An subset of a network” should be “A subset of a network”.
       --In Line157 of Page4, “The output output vector” should be “The output vector”.
       --In Line283 of Page7, “B0V2 as,” should be “B0V2 as teacher,”.
       --In Line301 of Page7, “due the inconsistencies” should be “due to the inconsistencies”.


**Summary Of The Paper:**

This paper proposes a neural network pruning and fine-tuning framework for model compression. It can automatically prune the channels by learning the channel importance. The contributions are: 1) A new pruning scheme is proposed by learning the channel importance; 2) The pruning logic is introduced in the pruning scheme. Thus, some grouping operations are pruned jointly. The pruning problems of residual connections in ResNet can be solved; 3) Hierarchical knowledge distillation is added in the fine-tuning phase to speed up training. Experimental results show effectiveness of the proposed method.

**Summary Of The Review:**

This paper proposes a new neural network pruning and fine-tuning framework for model compression. Each part of the proposed method is effective and can solve some challenges. However, the novelty and the technical details are limited, and the experimental results are not sufficient (see weakness). Therefore, my rating is marginally below the acceptance threshold.

---

### Official Review · Reviewer_b957 · 2021-11-02

**Correctness:** 3
**Technical Novelty And Significance:** 2
**Empirical Novelty And Significance:** 2
**Recommendation:** 3
**Confidence:** 3

**Main Review:**

#### Pros
- This paper aims at solving the coupled channels in channel pruning, which is important in practice and needs more investigations.
- The evaluated tasks (image denoising and human segmentation) seem to be quite interesting and seldomly discussed in related pruning works.
- It is good to observe and conclude that KD only works for pruned networks but not for unpruned (original) networks. However, more investigations are welcomed to make confident claims and hypotheses.

#### Cons
- There are only a few baseline pruning methods compared. Meanwhile, results on ResNet should be compared. Otherwise, it is difficult to judge the effectiveness of the proposed method.
- The tables show merely the same results as those in the figures (e.g., Table 1 & Figure 4, Tables 2/3 & Figure 5.
- No ablation studies. It seems that the most performance gain is from the hierarchical KD, which is not exclusive to other pruning methods. Applying hierarchical KD seems not to have enough methodological novelty or contributions.
- The writing clarity should be moderately addressed. For example, the descriptions of the two phases in Section 4.1 are quite informal and vague, and the relation between Algorithm 1 and the described steps in the main text is also unclear. Also, it is worthy to explain why "concatenation" is unique to others and important in your pruning method (if so).

#### Details
- It is not clear that how the proposed method, compared with baselines, uniquely solves the issue of pruning "specific numbers of input and output channels".
- The claim that the proposed method is "easy to scale and be deployed for segmentation, detection or image denoising" needs more explanations.
- It is not fully correct to claim "between 200M and 300M our pruned models outperform FBNetV2" from Figure 1 since FBNetV2 obtains an accuracy of ~76 with fewer FLOPs.
- For the four operation types introduced by the authors, which one does the "concatenation" belong to?

- Minor issues
  - In Abstract: "We release a set pruned ... " -> "a set of"
  - In Section 2: "Selecting channels based network ..." -> "based on"
  - In Section 2: "and train train our pruned network ..." -> "and train our pruned network"
  - In Section 3: "The output output vector of ..." -> "The output vector of"
  - Figure 2: "An subset of ..." -> "A subset"
  - In Section 3.1: "... a specific block size o channels ..." -> "of"
  - In Section 4.1: "In the fist phase" -> "first"

**Summary Of The Paper:**

This paper proposes a new channel pruning method, which firstly finds the coupled channels in the network, then prunes channels with learned logit predictors, and finally uses hierarchical knowledge distillation (KD) to fine-tune the pruned network. Empirical evaluations have been conducted with EfficientNets and MobileNets on ImageNet, an image denoising dataset, and a private human segmentation dataset.

**Summary Of The Review:**

This paper solves an important practical problem in channel pruning with interesting solutions and results. However, in terms of novelty, clarity, and evaluations, this paper needs to be polished and completed.

---

### Official Review · Reviewer_Gsnx · 2021-11-06

**Correctness:** 3
**Technical Novelty And Significance:** 3
**Empirical Novelty And Significance:** 3
**Recommendation:** 5
**Confidence:** 3

**Main Review:**

1) The research about channel pruning is insufficient, with only six related works, and the author should summarize the differences with them.
2) Hierarchical distillation of knowledge seems to be a traditional approach, which should not be a contribution to this work. In the experiments, the authors should verify the effectiveness of the proposed method by comparing it with various channel pruning and knowledge distillation methods.
3) The author should analyze the differences between the two parts, eg. New pruning algorithm and Hierarchical knowledge distillation, with other methods in the ablation study.

**Summary Of The Paper:**

This paper proposes a hierarchical knowledge distillation method for neural network pruning. Experiments demonstrate that the whole pruning pipeline requires much less computational resources than some of the state-of-the-art NAS based solutions for finding efficient FLOPs / accuracy trade-offs.

**Summary Of The Review:**

See the main review.

---

### Decision · Program_Chairs · 2022-01-20

**Decision:**

Reject

**Comment:**

A method for pruning neural networks is proposed.  Reviewers raised several concerns, including poor technical presentation and insufficient experimental validation with respect to both baseline methods and ablation studies.  All reviewer ratings lean toward reject and the authors did not provide a response.